# Low-dose theophylline in addition to ICS therapy in COPD patients: A systematic review and meta-analysis

**Tiankui Shuai[1,2], Chuchu Zhang[1,2], Meng Zhang[1,2], Yalei Wang[1,2], Huaiyu Xiong[1,2], Qiangru Huang[1,2], Jian Liu[1] ***

**1** Department of Intensive Care Unit, Lanzhou University, The First Hospital of Lanzhou University, Lan Zhou, Gansu Province, China, **2** The Fist Clinical Medical College of the First Hospital of Lanzhou University, Lanzhou University, Lanzhou, Gansu Province, China

* medecinliu@sina.com

**Data Availability Statement:** All relevant data are within the manuscript and its Supporting Information files.

## Abstract

### Background

A synergism has been reported between theophylline and corticosteroids, wherein theophylline increases and restores the anti-inflammatory effect of inhaled corticosteroids (ICS) by enhancing histone deacetylase-2 (HDAC) activity. Several studies have explored the efficacy of low-dose theophylline plus ICS therapy on chronic obstructive pulmonary disease (COPD) but the results are discrepant.

### Method

We conducted searches in electronic database such as PubMed, Web Of Science, Cochrane Library, and Embase to find out original studies. Stata/SE 15.0 was used to perform all data analysis.

### Result

A total of 47,556 participants from 7 studies were included in our analysis and the sample size of each study varied from 24 to 10,816. Theophylline as an add-on therapy to ICS was not associated with the reduction of COPD exacerbations (HR: 1.08, 95% CI: 0.97 to 1.19, $I^2$ = 95.2%). Instead, the theophylline group demonstrated a higher hospitalization rate (HR: 1.12, 95% CI: 1.10 to 1.15, $I^2$ = 20.4%) and mortality (HR: 1.19, 95% CI: 1.14 to 1.25, $I^2$ = 0%). Further, the anti-inflammatory effect of low-dose theophylline as an adjunct to ICS on COPD was controversial. Besides, the theophylline group showed significant improvement in lung function compared with the non-theophylline group.

### Conclusion

Based on current evidence, low-dose theophylline as add-on therapy to ICS did not reduce the exacerbation rate. Instead, the hospitalization rate and mortality increased with theophylline. Thus, we do not recommend adding low-dose theophylline to ICS therapy in COPD patients.

**Funding:** The authors received no specific funding for this work.

**Competing interests:** The authors have declared that no competing interests exist.

## Trial registration

**PROSPERO Registration** CRD42021224952.

## Introduction

Chronic obstructive pulmonary disease (COPD) is a common, preventable, and treatable chronic inflammatory lung disease that results in irreversible and progressive airflow limitation [1]. The airflow limitation is caused by significant exposure to noxious particles or gases [1]. COPD has now become one of the top three causes of death worldwide, and 90% of these deaths occur in low- and middle-income countries (LMICs) [2]. With aging of the population and continued exposure to COPD risk factors, the economic burden of COPD worldwide is projected to increase in the coming decades [3].

Theophylline is the most commonly used methylxanthine, a non-selective phosphodiesterase inhibitor, originally used as a bronchodilator in COPD [4]. Oral theophylline has been used as a bronchodilator to treat airway diseases for over 80 years and is currently widely used in resource-limited countries [5–7]. Because relatively high dose (10–20 mg/L) of theophylline are required and these are associated with frequent side effects [8], the Global Initiative for Chronic Obstructive Lung Disease (GOLD) guideline indicated that theophylline is not recommended in COPD patients unless other long-term treatment bronchodilators are unavailable or unaffordable [1].

Inhaled corticosteroids (ICS) have been used as an effective anti-inflammatory drug in chronic pulmonary inflammatory diseases, such as asthma [9]. Furthermore, ICS in combination with long-acting beta-2 agonists (LABA) have previously been recommended for moderate-to-severe airflow limitation in COPD patients in GOLD guideline [1, 10, 11]. However, evidence indicated that corticosteroids provided little clinical benefit and did not reduce the progression or mortality rate in COPD [9, 12]. The lack of response to corticosteroids in COPD may be associated with the reduction in the activity and expression of the critical enzyme histone deacetylase-2 (HDAC) activity as a result of increased oxidative stress [9, 13, 14].

Interestingly, synergism between theophylline and corticosteroids has been reported, wherein theophylline increases and restores the anti-inflammatory effects of ICS by enhancing HDAC activity [15–18]. It is worth noting that the abovementioned effect is achieved at a low plasma concentration of theophylline (1–5 mg/L) [16]. Furthermore, several randomized controlled trials and observational studies have explored the efficacy of low-dose theophylline added to ICS therapy in COPD, e.g., exacerbation frequency, lung function improvement, and changes in biomarkers [19–25].

However, the results of these studies were discrepant. Several studies reported that low-dose theophylline as add-on therapy to ICS did not enhance the anti-inflammatory effect of ICS and reduce COPD exacerbation frequency [19–21, 23]. In contrast, other research demonstrated that the addition of theophylline to ICS therapy was associated with the reduction of the exacerbation rate, improvement of lung function, and enhancement of anti-inflammatory effects [16, 22, 24]. Therefore, we conducted this meta-analysis to explore the efficacy and safety of adding theophylline to ICS therapy in COPD to provide reliable evidence for clinicians.

## Methods

All methods for conducting this systematic review and meta-analysis followed the PRISMA guidelines [25, 26]. The procedure is based on a protocol registered in the PROSPERO register of systematic reviews (CRD42021224952).

## Data source and searches

We conducted searches in electronic database such as PubMed, Web Of Science, Cochrane Library, and Embase from inception to October 31th, 2020 to find out original studies that described the efficacy of theophylline as add-on therapy to ICS on COPD patients. There was no languages restriction in our search process. We reviewed reference of all primary studies to make our search more comprehensive. When a duplicate publication of the same trial was found, the study with the most complete, recent, and updated report was included. The search was conducted with following keywords: theophylline, and ICS (beclomethasone, triamcinolone, flunisolide, budesonide or fluticasone) and chronic obstructive pulmonary disease. The detailed search strategies in databases are shown in supplementary.

Studies that met the following eligible criteria were included:

1. Studies that compared the efficacy between ICS plus theophylline therapy and without theophylline therapy in COPD patients.

2. Studies with subjects including individuals who had been predominantly diagnosed with COPD: a post-bronchodilator ratio of forced expiratory volume in the first second to forced vital capacity (FEV1/FVC) < 0.7.

3. Studies that reported at least one of the following outcomes: hazard ratio (HR) for exacerbation frequency, HR for hospitalization rate, HR for mortality, improvement of $FEV_1$, and changes in inflammatory or anti-inflammatory biomarkers (such as IL-6, IL-8, HDAC, TNF-α, and NFκB).

The exclusion criteria included the following:

1. Studies that used drugs with the potential to influence plasma theophylline concentration.

2. Studies that described the use of theophylline on other respiratory diseases, such as asthma.

3. Studies that included animal research, reviews, case reports, letters, and commentaries.

## Data extraction and quality assessment

Two authors (S.T.K and Z.C.C) reviewed the titles and/or abstracts of all retrieved studies independently, read the full text of included studies, and extracted data from original studies. We extracted the following data: first author, publication year, study design, location, sample size, mean age, gender, current smoking status, intervention, HR for exacerbation frequency, HR for hospitalizations, HR for mortality, improvement of FEV1, and changes in inflammatory or anti-inflammatory biomarkers. The primary outcome was HR for exacerbation frequency.

Two authors (Z.M and W.Y.L) individually performed the quality assessments. We used the Newcastle–Ottawa Scale (NOS) to assess the quality of the cohort studies [27], which contained three main concepts: selection, comparability, and outcome assessment. We characterized scores ≥7 as low risk of bias, 5–7 as moderate risk, and <5 as high risk. We assessed the methodology quality of randomized controlled trials based on Cochrane Handbook for Systematic Reviews of Interventions [28], which included six perspectives: random sequence generation (selection bias), allocation concealment (selection bias), blinding (performance bias and detection bias), incomplete outcome data (attrition bias), selective outcome reporting (attrition bias), and other potential sources of bias. Besides, the criteria for grading studies were as follows: (1) trials were graded as low quality if either randomization or allocation concealment was assessed to have a high risk of bias, regardless of other items; (2) trials were

graded as high quality when both randomization and allocation concealment were assessed to have a low risk of bias, and all other items were assessed to have a low or unclear risk of bias in a trial; (3) trials were graded as moderate quality if they did not meet the criteria for high or low risk. In case of any discrepancy, an agreement was reached through discussion among all authors.

## Data analysis

Stata/SE 15.0 was used to perform all extracted data. We pooled the adjusted HR and 95% CI to analyze the exacerbation rates, mortality, and hospitalization rate. Regarding the improvement of lung function and inflammatory biomarkers, we performed a systematic review because of lack of original data. Heterogeneity was assessed by the $I^2$ statistic versus the $P$-value. We considered $P$-value $\leq 0.05$ and $I^2 \geq 50\%$ as high heterogeneity; $I^2 \leq 50\%$ indicated heterogeneity in an acceptable range. In this case, we selected a fixed effect model to analyze data. Otherwise, a random effect model was chosen. We conducted sensitivity analysis to detect if the results were reliable and stable. Egger's Test and Begg's Test were used to assess publication bias [29]. We constructed a funnel plot when the studies were more than 10 [30]. $P < 0.05$ indicated statistical significance.

## Results

### Eligible studies and risk of bias

We finally obtained 4,010 studies from four databases and additional records identified through other sources. After removing duplication, there remained 3,671 studies. After screening the titles and abstracts, 17 studies were included. We reviewed the full texts of these 17 studies, and finally 7 studies fulfilled eligibility criteria. The detailed process for this is shown in Fig 1.

### Description of included studies

A total of 47,556 participants were included from 7 studies and the sample size for a single study ranged between 24 and 10,816. The characteristics of the included studies are shown in Table 1. In a single study, the proportion of males ranged from 53.7% to 100%, the proportion of participants who smoked ranged from 2.95% to 57.7%. Of the seven included records, four studies were RCTs and three were observational cohort studies. The results of quality assessment were as follows: NOS scores were ranged from 5 to 7 in three cohort studies. All cohort studies were at moderate risk. Of the four RCTs, two were high quality and two were moderate quality. The risk of bias in the six items of the Cochrane instrument are displayed in S1 and S2 Figs.

### Exacerbation rate of COPD

Five records described the HR of COPD exacerbations [19–21, 23, 25]. The meta-analysis result demonstrated that theophylline as an add-on therapy to ICS was not associated with the reduction of COPD exacerbation (HR: 1.08, 95%CI: 0.97 to 1.19, $I^2$ = 95.2%, Fig 2). In a subgroup analysis based on theophylline dose, the result demonstrated that high-dose theophylline led to a significant increase in COPD exacerbation (Fig 2). Apart from this, we conducted a subgroup analysis based on study design. RCTs and cohort studies both indicated that adding theophylline to ICS did not reduce COPD exacerbation (Fig 3).

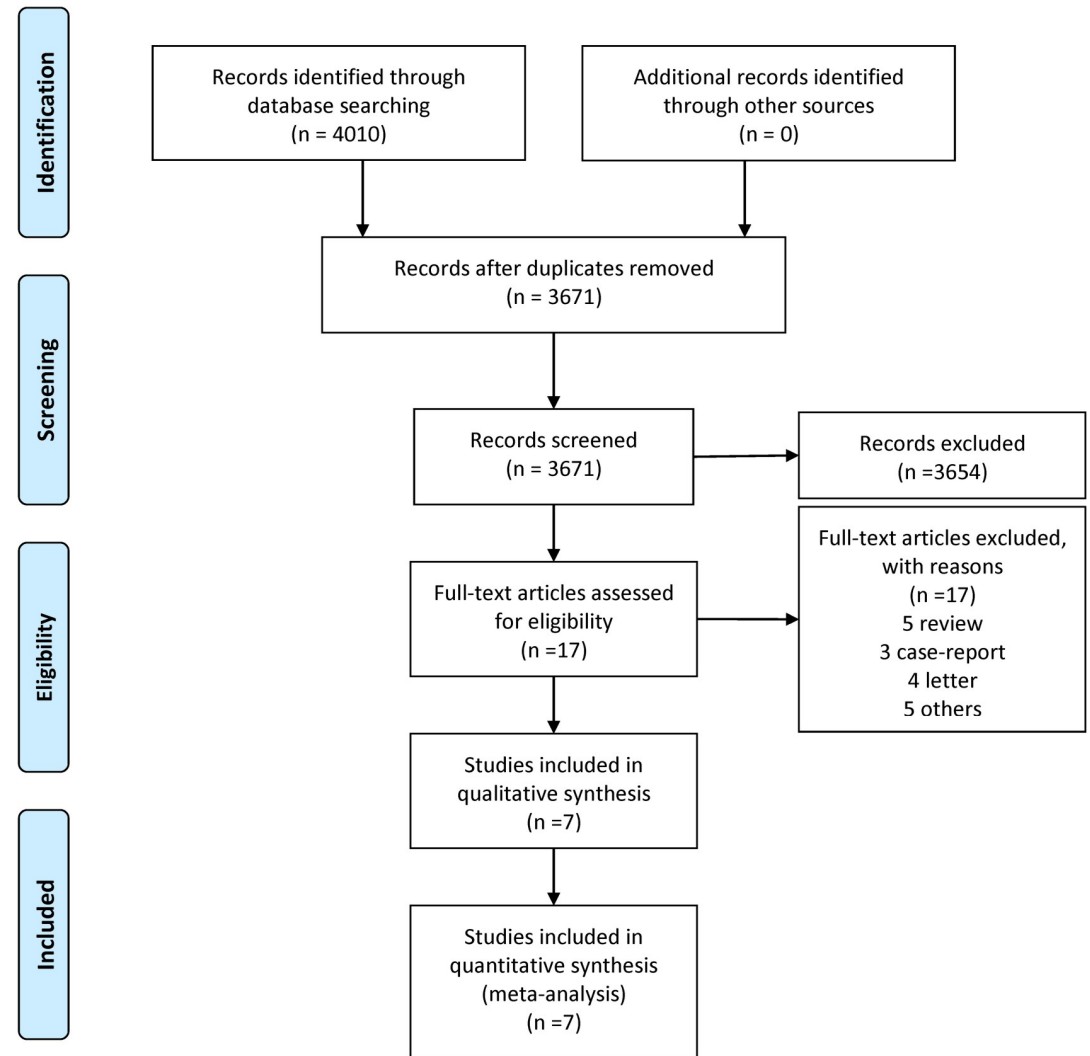

**Fig 1. Study selection process: PRISMA flow diagram identifying studies included in the meta-analysis.** Abbreviation: PRISMA, Preferred reporting Items for systematic reviews and Meta-analyses.

## Hospitalization rate and mortality of COPD

In this meta-analysis, the theophylline group demonstrated a higher hospitalization rate compared with the non-theophylline group [19, 20, 23, 24] (HR: 1.12, 95%CI: 1.10 to 1.15, $I^2$ = 20.4%, Fig 4). Similarly, the theophylline group was associated with an increased mortality of COPD patients compared with the non-theophylline group [20, 23] (HR: 1.19, 95%CI: 1.14 to 1.25, $I^2$ = 0%, Fig 5).

**Lung function and inflammatory biomarkers.** Lee et al. conducted a retrospective cohort study to explore the anti-inflammatory effect of low-dose theophylline as an adjunct to ICS in COPD patients [23]. The results described that the theophylline arm was associated with a statistically significant increase in HDAC activity and a further reduction in TNF-α and IL-8 concentrations in the sputum compared with the non-theophylline arm. However, a double-blind RCT by Cosio et al. reported that the HDAC activity and inflammatory biomarkers were not different in both groups either at baseline or at the end of the study [21]. Besides, Subramanian et al. conducted a single-blinded RCT to assess the safety profile of theophylline as

**Table 1. Characteristic of included studies (n = 7).**

| Study | Year | Design | N | Age | Male% | Smoker % | Duration | Intervention | Dosage(T) | NOS | outcome |
|---|---|---|---|---|---|---|---|---|---|---|---|
| Cyr, M.C. | 2007 | Cohort | 21760/10697 | 72.5±7.9/71.2±7.9 | 66.7/65.1 | NA | 172±269/185±237 days | T+ICS/LABA+ICS | 346±204 mg | 7 | ①② |
| Cosio, B.G. | 2009 | RCT | 16/19 | 67.6±1.3/66.7±1.7 | 100/0 | NA | 3 months | T/ST | 100mg bid | NA | ④ |
| Lee, T.A. | 2009 | Cohort | 1850/10816 | 71.4/69.0 | 94.0/91.5 | NA | 2002.10–2003.3 | a.T+ICS/ICS; b.T+ICS+LABA/ICS+LABA; c.T+ICS+SABA/ICS+LABA; d.T+ICS+LABA+SABA/ICS+LABA+SABA | 10–20 µg/ml | 5 | ①②③④ |
| Subramanian | 2015 | RCT | 24/26 | 57.96 ± 7.47/54.46 ± 10.49 | 87.5/96.2 | 50/57.7 | 60 days | T+ICS+LABA/ICS+LABA | > 50 kg: 400 mg; 40–50 kg: 300 mg; < 40 kg: 200 mg qd | NA | ⑤ |
| Cosio, B.G. | 2016 | RCT | 34/36 | 68.09 ± 8.37/67.82 ± 9.34 | 83.3/79.4 | 32.4/36.1 | 52 weeks | T+ICS+LABA/ICS+LABA | 100mg bid | NA | ①④ |
| Devereux, G. | 2018 | RCT | 788/779 | 68.3 ±8.2/68.5±8.6 | 53.9/53.7 | 31.4/32.0 | 52 weeks | T+ICS/ICS | 200mg qd or bid | NA | ①②③ |
| Wilairat, P. | 2019 | Cohort | 474/237 | 70.02 ±10.68/70.29±11.41 | 73.84/75.53 | 2.95/6.33 | 2011.1–2015.12 | T+ICS+LABA/ICS+LABA | ≤200mg qd or >200mg qd | 6 | ①② |

Outcome:①exacerbation rate;②hospitalization rate;③mortality;④FEV1;⑤HDAC or inflammatory biomarkers. Abbreviations: T: theophylline; ICS: Inhaled corticosteroids; LABA: long-acting beta-2 agonists; ST: standard therapy; IPR: ipratropium; PBO: placebo; NA: not applicable; NOS: Newcastle-Ottawa Scale.

an adjunct to ICS in COPD [22]. This study demonstrated that the theophylline group showed significant improvement in $FEV_1$.

## Sensitivity analysis and publication bias

After sensitivity analysis, we observed the overall findings remained consistent. We used funnel plots to access the publication bias (S3 Fig), and these results did not show any evidence of obvious bias (Egger's test, $P = 0.419$, S4 Fig).

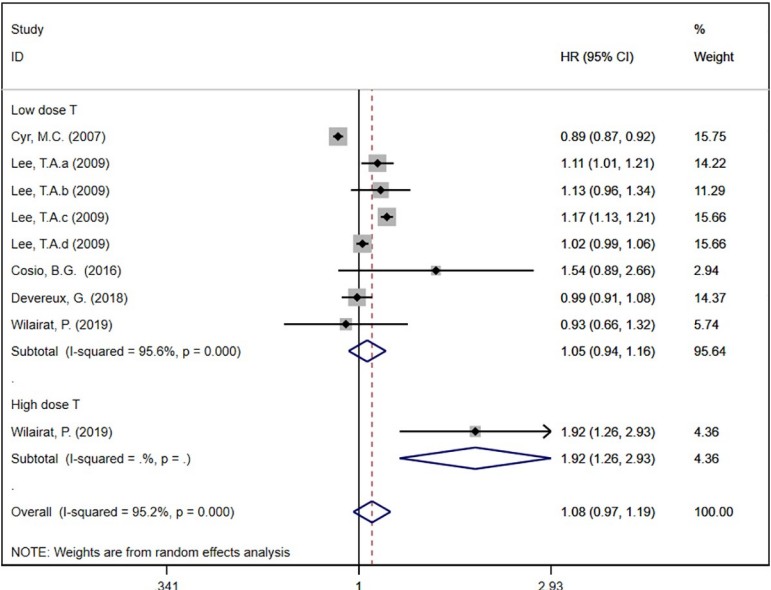

**Fig 2. Forest plot of acute exacerbation rate (Subgroup analysis based on the dose of theophylline).**

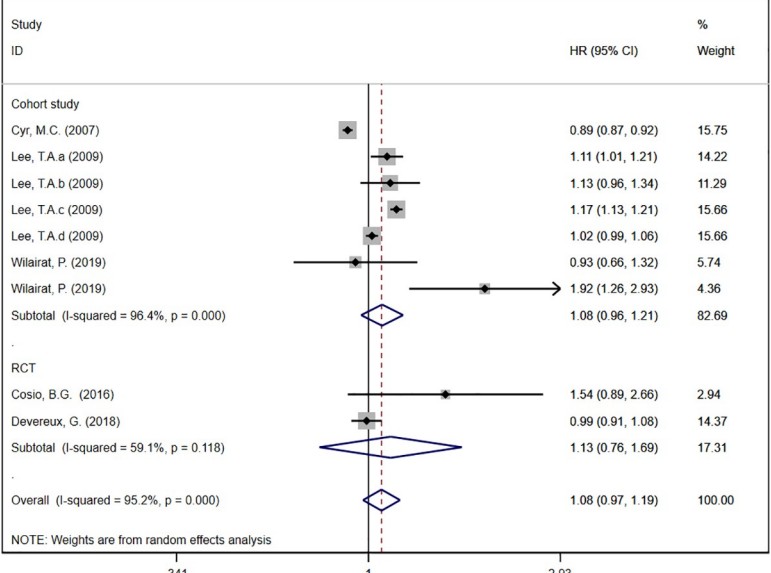

**Fig 3. Forest plot of acute exacerbation rate (Subgroup analysis based on study design).**

## Discussion

This meta-analysis demonstrated the efficacy and safety of theophylline as an add-on therapy to ICS in COPD. We found that theophylline was not associated with a reduction of exacerbation rates. Instead, the hospitalization rates, and mortality of COPD patients increased with the use of theophylline. Besides, the anti-inflammatory effect of theophylline on COPD in original studies was inconsistent. Furthermore, there was a study that indicated that the use of theophylline as an add-on therapy to ICS improved lung function in patients with COPD [22]. Overall, the findings of this meta-analysis do not support the use of theophylline as an adjunctive therapy to ICS treatment for COPD patients.

This meta-analysis indicated theophylline as an adjunct to ICS did not reduce exacerbation. Instead, the hospitalization rate and mortality of COPD patients increased. Likewise, Horita N

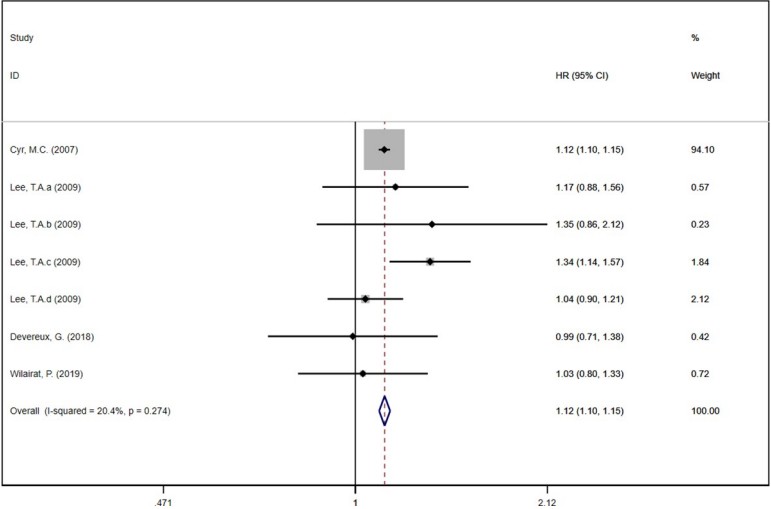

**Fig 4. Forest plot of hospitalization rate.**

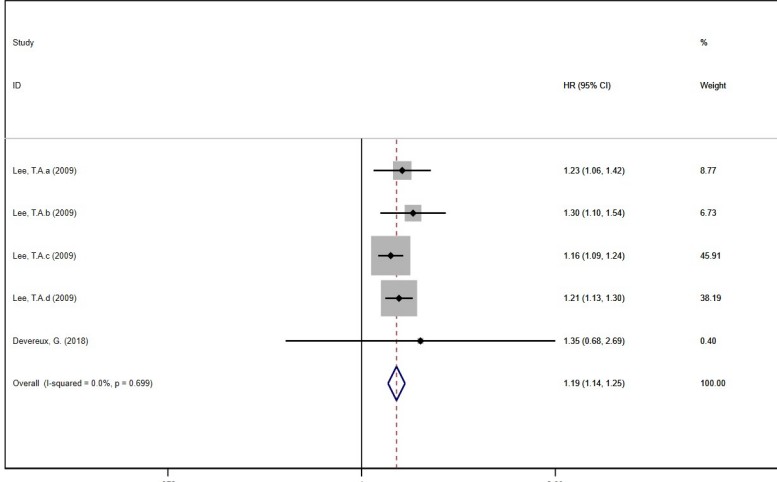

**Fig 5. Forest plot of mortality.**

et al. conducted a meta-analysis to explore the impact of theophylline on mortality in COPD patients [31]. The study found that theophylline slightly increased all-cause mortality in COPD patients. In contrast, there was a double-blind, parallel-group, placebo-controlled RCT that evaluated the therapeutic effect of low-dose theophylline treatment (100 mg twice daily) [32]. This one-year study demonstrated a different result reporting that theophylline reduced the COPD exacerbation frequency ($P = 0.047$ and $P = 0.035$, respectively). Importantly, theophylline was considered to be an anti-inflammatory agent apart from being a bronchodilator [33].

Regarding lung function, the results of our systematic review demonstrated that the theophylline group showed significant improvement in FEV1 compared with the ICS plus LABA group in COPD. Coincidently, Broseghini C et al. studied the efficacy of theophylline compared with LABA or placebo and reported that theophylline improved significantly, but less, the $FEV_1$ (about 80 ml, on average) without affecting any of the other lung function variables [34]. Another systematic review and meta-analysis found that theophylline treatment improved $FEV_1$ with a weighted mean difference of 100 mL compared with the placebo [4]. Rossi A et al. also indicated that the effectiveness of theophylline in lung function improvement was less than that of LABA [35]. Additionally, reports indicating adverse events following the use of theophylline such as palpitation, tremor, and other arrhythmias were frequently reported [8, 36–38]. Thus, the effect of theophylline on improving lung function was limited. Because of the adverse events associated with theophylline use, it should be cautiously prescribed clinically.

Furthermore, the anti-inflammatory effect of theophylline as an additional therapy to ICS in COPD is debated in this systematic review and meta-analysis. Theoretically, theophylline inhibits phosphodiesterase [39] and other inflammatory mediators [40], increases apoptosis [41], and inhibits NF-κB [42] at higher concentrations than those used in practice (>20 mg/ L). At lower doses, theophylline increases HDAC2 activity by inhibiting phosphoinositide-3-kinase-δ (PI3Kδ) [15, 17], which is activated by oxidative stress [17]; reduces neutrophil concentration in the sputum [43, 44] and large airways [45]; and enhances the anti-inflammatory effects of glucocorticoids [46]. The possible explanation for why theophylline did not increase HDAC activity and reduce the inflammatory biomarker levels were as follows. First, there was a lack of *in vivo* biological effect of low-dose theophylline treatment. Second, theophylline level

was too low to achieve an effect. Finally, the anti-inflammatory effect did not sustain in the long term.

The heterogeneity was high for the exacerbation rate. A subgroup analysis for detecting the source of heterogeneity identified the study design as the source of heterogeneity. We considered the heterogeneity was because of several reasons. First, the exact dosage of theophylline was different in the included studies and the dosage may influence the pharmacological effect of theophylline *in vivo*. Second, the intervention following in the exposure group of included studies was not exactly the same; other than theophylline and ICS, LABA and ipratropium were also included in some studies. The difference in interventions may have caused the different effect on COPD patients. Accordingly, this could be a possible source of high heterogeneity. Third, the duration of study varied from 60 days to 4 years. We believe that the duration may influence the effect of theophylline on COPD; accordingly, it was also regarded as a source of heterogeneity.

This systematic review and meta-analysis have several limitations. First, because of a lack of original studies, we conducted a systematic review for inflammatory biomarkers and $FEV_1$, which may have reduced the reliability of results inevitably. Thus, further studies to support our conjecture are needed in the future. Second, although we conducted a comprehensive search to identify as many studies as possible, the number of studies eventually included in the analysis was still small and the sample size was insufficient; this may have reduced the generalizability of our meta-analysis results. Finally, the quality of life for COPD patients was not explored based on the existing data. This needs to be analyzed in the future.

## Conclusion

In this systematic review and meta-analysis, low-dose theophylline as an add-on therapy to ICS did not reduce the exacerbation rate of COPD. Instead, the hospitalization rate and mortality increased. There was a controversy concerning the anti-inflammatory effect of low-dose theophylline. Furthermore, theophylline as an add-on therapy to ICS improved lung function compared with non-theophylline group. Thus, we do not recommend adding low-dose theophylline to ICS therapy in COPD patients based on current evidence.

## Supporting information

**S1 Fig. Risk of bias graph presenting each risk of bias item as percentages across all included studies.**
(TIF)

**S2 Fig. Risk of bias summary for included studies, showing each risk of bias item for every included study.**
(TIF)

**S3 Fig. Funnel plot for publication bias.**
(TIF)

**S4 Fig. Egger's publication bias.**
(TIF)

**S1 File. The detailed search strategy.**
(DOC)

**S2 File. PRISMA 2009 checklist.**
(DOC)

**S3 File. PROSPERO protocol.**
(PDF)

**S4 File. Certificate of editing.**
(PDF)

## Acknowledgments

We are very grateful for the sharing of results provided by the authors of the included studies.

## Author Contributions

**Conceptualization:** Tiankui Shuai, Jian Liu.

**Methodology:** Yalei Wang, Huaiyu Xiong, Qiangru Huang.

**Software:** Meng Zhang, Huaiyu Xiong, Qiangru Huang.

**Writing – original draft:** Tiankui Shuai, Chuchu Zhang.

**Writing – review & editing:** Tiankui Shuai, Chuchu Zhang.

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
