## [Decision Letter · Decision Letter 0]

12 Feb 2021

PONE-D-21-02298

Adding Low-dose Theophylline to ICS therapy on COPD: a systematic review and meta-analysis

PLOS ONE

Dear Dr. Liu,

Thank you for submitting your manuscript to PLOS ONE. After careful consideration, we feel that it has merit but does not fully meet PLOS ONE’s publication criteria as it currently stands. Therefore, we invite you to submit a revised version of the manuscript that addresses the points raised during the review process.

We look forward to receiving your revised manuscript.

Kind regards,

Walid Kamal Abdelbasset, Ph.D.

Academic Editor

PLOS ONE

Journal Requirements:

Reviewers' comments:

Reviewer's Responses to Questions

**Comments to the Author**

1. Is the manuscript technically sound, and do the data support the conclusions?

Reviewer #1: Partly

Reviewer #2: Partly

2. Has the statistical analysis been performed appropriately and rigorously? 

Reviewer #1: Yes

Reviewer #2: N/A

3. Have the authors made all data underlying the findings in their manuscript fully available?

Reviewer #1: No

Reviewer #2: Yes

4. Is the manuscript presented in an intelligible fashion and written in standard English?

Reviewer #1: No

Reviewer #2: Yes

5. Review Comments to the Author

Reviewer #1: Paper titled (Adding Low-dose Theophylline to ICS therapy on COPD: a systematic review and meta-analysis) discussed the valued of adding theophylline to the traditional ICS therapy to COPD patients. ALthough the manuscript has merit but many recommendations can be follwoed before consideration for publication.

Title: on COPD should be in COPD patients or cases

Abstract: man abbreviations appeared without definition at the first appearance. I believe Plos one is a multidisciplinary journal and clarifications of these terms before use is mandatory.

Abstract: first line (It has been reported that there was synergism...) can be replaced by (a synergism has been reported)''

The conclusion is not useful (by the way it is exactly the same as that written after discussion) and cannot give the reader what was the outcome of using theophylline whether good or bad. whether will be recommended or not

Also (There were several studies explored) can be , Sevral studies explored............

Kindly revise the whole manuscript for similar non-perfect terms and sentences. I recommend if authors seek help from a naive speaker or editing services.

Resolution of all images should be 300 dpi as minimal (or better 600 dpi).

Reviewer #2: 1. Line 132

Studies that did not exclude patients who used drugs that interact with theophylline should be excluded.

2. Lines 126, 127

Please check (The subjects included who had been predominantly diagnosed with COPD: ratio of forced expiratory volume in the first second to forced vital capacity (FEV1/FVC) <0.7) as the presence of post bronchodilator (FEV1/FVC) <0.7 confirms the presence of airflow limitation and COPD diagnosis.

3. Table (1)

There is no equality in intervention therapies between selected studies as well as study of 2007 did not mention the used standard therapy which makes your hypothesis not accurate.

4. Selected studies did not afford data about exact dosage of theophylline and the dosage might influence the pharmacological action of theophylline .Also there is no available data about duration of treatment as variability in dose and duration increase heterogeneity.

5. Line 242

Select a study you refer in this sentence (there was a study that indicated

Theophylline as add-on therapy to ICS improved lung function of COPD).

6. PLOS authors have the option to publish the peer review history of their article (what does this mean?). If published, this will include your full peer review and any attached files.

Reviewer #1: **Yes: **Sawsan A. Zaitone

Reviewer #2: No

---

## [Author Response · Author response to Decision Letter 0]

28 Mar 2021

We have studied the valuable comments from you, the assistant editor and reviewers carefully, and tried our best to revise the manuscript. The point-to-point response to the reviewers and editor are listed as following:

Respond to reviewer’s comments:

Reviewer#1:

Paper titled (Adding Low-dose Theophylline to ICS therapy on COPD: a systematic review and meta-analysis) discussed the valued of adding theophylline to the traditional ICS therapy to COPD patients. Although the manuscript has merit but many recommendations can be followed before consideration for publication.

Title: on COPD should be in COPD patients or cases

Thanks for the reviewer’s suggestion. The title of the article has been revised as following: “Low-dose theophylline in addition to ICS therapy in COPD patients: a systematic review and meta-analysis”. I have corrected it in manuscript and highlighted with yellow mark in the revised manuscript with track changes (supporting information*). 

Abstract: many abbreviations appeared without definition at the first appearance. I believe Plos one is a multidisciplinary journal and clarifications of these terms before use is mandatory.

Thanks for the reviewer’s suggestion. I have added the definitions of abbreviations at the first appearance in abstract and highlighted with yellow mark in the revised manuscript with track changes (supporting information*).

Abstract: first line (It has been reported that there was synergism...) can be replaced by (a synergism has been reported)

Thanks for the reviewer’s suggestion. The corresponding part of the article has been revised as following: “A synergism has been reported between theophylline and corticosteroids, wherein theophylline increases and restores the anti-inflammatory effect of inhaled corticosteroids (ICS) by enhancing histone deacetylase-2 (HDAC) activity”. I have revised it in manuscript and highlighted with yellow mark in the revised manuscript with track changes (supporting information*).

The conclusion is not useful (by the way it is exactly the same as that written after discussion) and cannot give the reader what was the outcome of using theophylline whether good or bad. whether will be recommended or not

Thanks for the reviewer’s suggestion. The corresponding part of the article has been revised as following: “Based on current evidence, low-dose theophylline as add-on therapy to ICS did not reduce the exacerbation rate. Instead, the hospitalization rate and mortality increased with theophylline. Thus, we do not recommend adding low-dose theophylline to ICS therapy in COPD patients”. I have revised it in manuscript and highlighted with yellow mark in the revised manuscript with track changes (supporting information*).

Also (There were several studies explored) can be , Several studies explored............

Thanks for the reviewer’s suggestion. The corresponding part of the article has been revised as following: “Several studies have explored the efficacy of low-dose theophylline plus ICS therapy on chronic obstructive pulmonary disease (COPD) but the results are discrepant”. I have corrected it in manuscript and highlighted with yellow mark in the revised manuscript with track changes (supporting information*).

Kindly revise the whole manuscript for similar non-perfect terms and sentences. I recommend if authors seek help from a naive speaker or editing services.

Thanks for reviewer’s comment. I have invited a native English-editor to help me correct the grammar and style errors. The version with track changes of editing agency and the certification of editing has been uploaded to the journal.

Resolution of all images should be 300 dpi as minimal (or better 600 dpi).

Thanks for reviewer’s comment. I have corrected the resolution of the image and uploaded the correct version.

Reviewer #2: 

1. Line 132 Studies that did not exclude patients who used drugs that interact with theophylline should be excluded.

Thanks for reviewer’s comment. I have added one exclusion criteria as following: “studies that used drugs with the potential to influence plasma theophylline concentration”. Also, I have rechecked the included studies, none of which met the exclusion criteria.

2. Lines 126, 127 Please check (The subjects included who had been predominantly diagnosed with COPD: ratio of forced expiratory volume in the first second to forced vital capacity (FEV1/FVC) <0.7) as the presence of post bronchodilator (FEV1/FVC) <0.7 confirms the presence of airflow limitation and COPD diagnosis.

Thanks for reviewer’s comment. The corresponding part of the article has been revised as following: “Studies with subjects including individuals who had been predominantly diagnosed with COPD: a post-bronchodilator ratio of forced expiratory volume in the first second to forced vital capacity (FEV1/FVC) < 0.7”. I have corrected it in manuscript and highlighted with yellow mark in the revised manuscript with track changes (supporting information*).

3. Table (1) There is no equality in intervention therapies between selected studies as well as study of 2007 did not mention the used standard therapy which makes your hypothesis not accurate.

Thanks for reviewer’s comment. The included studies in this meta-analysis did exist inconsistencies in the basic therapy regimens. And we listed the detailed medication of each studies in table1 (“Intervention”). Also, we tried to perform subgroup analysis based on medication type. But there were so few studies of each treatment regimen that there is only one study in a subgroup. Therefore, the subgroup analysis did not address the issue very well. This is indeed a source of heterogeneity for our meta-analysis. Thus, we describe it in the discussion section (line 289-293): “the intervention following in the exposure group of included studies was not exactly the same; other than theophylline and ICS, LABA and ipratropium were also included in some studies. The difference in interventions may have caused the different effect on COPD patients. Accordingly, this could be a possible source of high heterogeneity”. 

4. Selected studies did not afford data about exact dosage of theophylline and the dosage might influence the pharmacological action of theophylline. Also, there is no available data about duration of treatment as variability in dose and duration increase heterogeneity.

Thanks for reviewer’s comment. The included studies provided the specific dosage or the dosage ranges of the theophylline. Based on the information from included researches, we added the dosage of theophylline to the Table 1. And we conducted the subgroup analysis based on the theophylline dosage (Fig 2), indicating that the dosage might be a source of heterogeneity. Furthermore, we have described it in the discussion section (line 288, 289): “the exact dosage of theophylline was different in the included studies and the dosage may influence the pharmacological effect of theophylline in vivo”. Meanwhile, as the reviewer considered, the included studies in this meta-analysis did exist inconsistencies in the duration. And we listed the duration of each studies in table1 (“Duration”). Considering the dose-time-effect, individual differences, and the fact that most studies did not continuously monitor theophylline concentrations in plasma, we believe it is also part of the source of heterogeneity. we have described it in the discussion section (line 294-296): “the duration of study varied from 60 days to 4 years. We believe that the duration may influence the effect of theophylline on COPD; accordingly, it was also regarded as a source of heterogeneity”. 

5. Line 242 Select a study you refer in this sentence (there was a study that indicated Theophylline as add-on therapy to ICS improved lung function of COPD).

Thanks for reviewer’s comment. I have added the study in manuscript and highlighted with yellow mark in the revised manuscript with track changes (supporting information*).

---

## [Decision Letter · Decision Letter 1]

26 Apr 2021

Low-dose theophylline in addition to ICS therapy in COPD patients: a systematic review and meta-analysis

PONE-D-21-02298R1

Dear Dr. Liu,

We’re pleased to inform you that your manuscript has been judged scientifically suitable for publication and will be formally accepted for publication once it meets all outstanding technical requirements.

Kind regards,

Walid Kamal Abdelbasset, Ph.D.

Academic Editor

PLOS ONE

Additional Editor Comments (optional):

Reviewers' comments:

Reviewer's Responses to Questions

**Comments to the Author**

1. If the authors have adequately addressed your comments raised in a previous round of review and you feel that this manuscript is now acceptable for publication, you may indicate that here to bypass the “Comments to the Author” section, enter your conflict of interest statement in the “Confidential to Editor” section, and submit your "Accept" recommendation.

Reviewer #1: All comments have been addressed

Reviewer #3: All comments have been addressed

2. Is the manuscript technically sound, and do the data support the conclusions?

Reviewer #1: Yes

Reviewer #3: Yes

3. Has the statistical analysis been performed appropriately and rigorously? 

Reviewer #1: Yes

Reviewer #3: Yes

4. Have the authors made all data underlying the findings in their manuscript fully available?

Reviewer #1: Yes

Reviewer #3: Yes

5. Is the manuscript presented in an intelligible fashion and written in standard English?

Reviewer #1: Yes

Reviewer #3: Yes

6. Review Comments to the Author

Reviewer #1: The revised version of Paper titled (Low-dose theophylline in addition to ICS therapy in COPD patients: a systematic

review and meta-analysis) was adequately revised by the authors.

Thanks for the authors for addressing the recommendations.

Reviewer #3: (No Response)

7. PLOS authors have the option to publish the peer review history of their article (what does this mean?). If published, this will include your full peer review and any attached files.

Reviewer #1: **Yes: **Sawsan A. Zaitone

Reviewer #3: No

---

## [Editor Report · Acceptance letter]

14 May 2021

PONE-D-21-02298R1 

Low-dose theophylline in addition to ICS therapy in COPD patients: a systematic review and meta-analysis 

Dear Dr. Liu:

I'm pleased to inform you that your manuscript has been deemed suitable for publication in PLOS ONE. Congratulations! Your manuscript is now with our production department. 

Kind regards, 

on behalf of

Dr. Walid Kamal Abdelbasset 

Academic Editor

PLOS ONE